# DEEP MULTIPLE INSTANCE LEARNING WITH GAUSSIAN WEIGHTING

## ABSTRACT

In this paper we present a deep Multiple Instance Learning (MIL) method that can be trained end-to-end to perform classification from weak supervision. Our MIL method is implemented as a two stream neural network, specialized in tasks of instance classification and weighting. Our instance weighting stream makes use of Gaussian radial basis function to normalize the instance weights by comparing instances locally within the bag and globally across bags. The final classification score of the bag is an aggregate of all instance classification scores. The instance representation is shared by both instance classification and weighting streams. The Gaussian instance weighting allows us to regularize the representation learning of instances such that all positive instances to be closer to each other w.r.t. the instance weighting function. We evaluate our method on five standard MIL datasets and show that our method outperforms other MIL methods. We also evaluate our model on two datasets where all models are trained end-to-end. Our method obtain better bag-classification and instance classification results on these datasets. We conduct extensive experiments to investigate the robustness of the proposed model and obtain interesting insights.

## 1 INTRODUCTION

Multiple instance learning (MIL) [7] is a form of weakly supervised learning that have been widely applied in drug activity prediction [7], medical imaging [14], text analysis and natural language processing [2] and computer vision [18]. In contrast to the standard supervised learning where every training instance is provided with a label, in MIL, training instances are grouped in sets, called *bags*, and labels are provided at bag-level (i.e. one label for entire bag). In the case of binary classification, a bag is labeled as positive, if there is at-least one positive sample, otherwise it is negative. As the instance labels are only indirectly accessible through labels attached to bags, the ambiguity of not knowing which instances are positive makes the problem significantly more challenging than the standard supervised classification. As the ratio of negatives to the positives increases in bags, the ambiguity increases and this renders the problem of bag classification harder. Our hypothesis is that the ability of robustly detecting positive instances in a bag is essential for successful MIL methods.

There is a rich body of literature in MIL which can broadly be grouped in two main categories, bag-level [17] and instance-level approaches [1, 22]. The first group maps instances of a bag into a single embedding (e.g. Fisher Vector [17]) and trains bag-level predictors to classify each bag in a similar manner to the standard classification. The second approach learns instance-level classifiers and aggregates the scores of individual instances from each bag. Though the former can be thought to provide a more unbiased solution to the problem of bag classification, the instance-level approach has recently been shown to outperform the bag-level one in [16, 10] under the assumption that bag labels can be predicted from individual instances. However, predicting instance labels from bag-level ones is an ill-posed problem. Thus, previous work proposes constrained solutions to tackle the ambiguity. Liu et al. [13] propose a voting framework to capture neighborhood relationship between instances from different bags to identify key instances. To this end, the authors use the voting framework to obtain a weighted nearest neighbor (NN) classifier where its weights globally generated from a kNN graph. Conditional random fields (CRF) are used to model relationships between bags in [6]. This models bags as nodes in a CRF with instances as their states. However, these methods are not learned end-to-end and thus relies on the feature space metric to obtain meaningful weights for each instance.

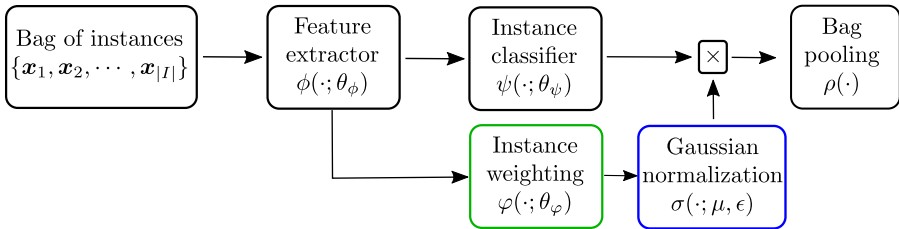

Figure 1: Our MIL model. First, we extract features from a deep feature extractor $\phi$ for each instance. Then the instance classifier $\psi$ outputs classification scores for each instance. The bottom stream consists of two sub-functions, the instance weighting $\varphi$ and the Gaussian normalization function $\sigma$. Finally, the scores of two streams are multiplied and aggregated by $\rho$ to get the bag score.

Deep neural networks are also widely applied to solve MIL problem [15, 22, 16]. Zhou et al. [22] aggregate the outputs of instance level classifier by the max operator to ensure that if at least one instance of a positive training bag is predicted as positive, then the concerned bag is assumed as positive. Similarly, if all instances of a negative bag are predicted as negative then the bag is negative. However, this method does not explicitly model for identifying the key instances in the neural network architecture but rather hope that feature representation of the instances would adapt (network would learn) such that instance level classifier can solely learn to give higher scores for positive instances from this loss function. Ramon and De Raedt [15] introduce log-sum-exp as the convex max to calculate bag probabilities from instance probabilities. Wang et al. [16] introduce differential MIL pooling layer to enforce MIL constraints such as that each positive bag should have at-least one positive instance. Recently, attention-based deep multiple instance learning was proposed where attention is applied over the features using Softmax attention before obtaining a bag level representation [10].

We also build on the idea of instance level classification, however, we take a different approach from all these methods in three aspects: First, we propose to take a two stream approach for instance classification and instance weighting (see fig. 1). Specifically, we have two dedicated paths or streams in our neural network, one dedicated for instance classification by returning a score and the other stream dedicated for learning normalized weights. We show empirically that this is crucial. Second, we do not specifically enforce any MIL constraint (e.g. number of positives in bag, certain graph structure) but rather choose a Gaussian activation function that allows us to learn a soft constraint to weight instances by comparing instances within bags and/or across bags. We investigate two variants of Gaussian activated weighting functions that employ bag-level or cross-bag statistics to assign weights to each instance. By comparing instances within a bag or across bags, our Gaussian activated weighting function learns meaningful weights to detect key instances within a bag. It assigns a higher weight for positive instances while assigns a small value for negative instances. In-fact, the instance weighing function assigns lower weights to the instances that are far from the mean and thus encourages positive instances to form compact clusters in the feature space. Enforcing similarity among the features of positive instances through a single mean per class also has a strong regularization on the feature space which implicitly enforces constraints on MIL. Third, we simply extend our binary MIL method to a multi-class bag-classification and our method is robust against severe ambiguity levels (very few positives in a very large bag). We demonstrate that our method obtain state-of-the-art results in the standard MIL instance learning benchmarks.

## 2 METHOD

**Problem definition.** In standard classification, it is assumed that we are given a set of training pairs $(\boldsymbol{x}, y) \in \mathcal{R}^d \times \mathcal{Y}$ with the goal of learning a predictor from inputs to labels $f : \mathcal{R}^d \to \mathcal{Y}$. In this paper, we focus on the more general problem of MIL and use a weaker form of supervision. Input samples are now organized into bags $\boldsymbol{B}_1, \boldsymbol{B}_2, ..., \boldsymbol{B}_m$ with each bag contains a variable number of samples, i.e. $\boldsymbol{B}_I = \{\boldsymbol{x}_i : i \in \{1, \cdots, |I|\}$ where $|I|$ is the number of instances in $\boldsymbol{B}_I$. In this setting, individual instances do not have access to a label but each bag $\boldsymbol{B}_I$ is associated with a label $Y_I \in \mathcal{Y}$. We consider two settings, binary and multi-class classification. In the binary case $\mathcal{Y} = \{-1, 1\}$, if a bag is negative, all instances in the bag are negative. On the other hand, if a bag is positive, at least one of the instances is positive. In the multi-class case, the output space

is categorical, i.e. $\mathcal{Y} = \{1, 2, \cdots, C\}$ where $C$ is the number of categories. We learn a classifier $g : \mathcal{X}^{|I|} \to \mathcal{Y}$ that takes in a bag of instances and outputs high score only for the target class:

$$g(\boldsymbol{B}_I) = \rho(f(\boldsymbol{x}_1), \cdots, f(\boldsymbol{x}_{|I|})) \qquad (1)$$

where $\rho$ is a symmetric function as in [19, 10, 4] (i.e. permutation invariant to the ordering of instances $\boldsymbol{x}$ in a bag) which can be decomposed into predictions for individiual instances $f(\boldsymbol{x})$. In other words, as we do not have access to ground-truth labels $y$ for individual samples $\boldsymbol{x}$, instance level predictions $f(\boldsymbol{x})$ are aggregated to obtain bag-level prediction $g(\boldsymbol{B})$. Examples of the aggregation function $\rho$ are *max* operator [1], *weighted sum* operator [10] and *log of sum of exponentials* [15]. As a matter of fact, we also build our MIL method on an aggregation function takes in instance level predictions, predicts the importance of each and aggregates them by taking a weighted sum to obtain a bag level prediction. In particular, we extend the idea present in eq. (1) to weight each instance using a neural network specifically designed to select *key instances* [13] i.e. discriminative instances that can reliably be utilized to predict bag labels. Formally we define each key instance to be positive below:

**Definition.** *Given a bag of multiple instances, if the bag label is positive, the key instances are the instances having positive label.*

**Property.** *Given a bag of multiple instances, if the bag is positive, key instances are fewer than non-key instances.* A diagram of our model is depicted in fig. 1 which consists of two streams, each stream contains multiple components. We describe them in the following.

**Instance classifier $f$.** First we define the instance level prediction $f(\boldsymbol{x}) = \psi_{\theta_\psi} \circ \phi_{\theta_\phi}$ as a composition of a feature extractor $\phi : \mathcal{X} \to \mathcal{H}$ where $\mathcal{H}$ is latent variable space and of a classifier $\psi$ where $\psi(\phi(x; \theta_\phi); \theta_\psi) \in R^C$ and $C$ is the number of target categories. In fact, both $\phi$ and $\psi$ are modeled as deep neural networks with learnable parameters $\theta_\phi$ and $\theta_\psi$ respectively.

**Instance weighting $\beta$.** The bag classifier $g(\boldsymbol{B}_I)$ takes in the instance scores $f(\boldsymbol{x}_1), \cdots, f(\boldsymbol{x}_{|I|})$ for $\boldsymbol{B}_I$ and outputs a weighted sum of them by using $\beta(\cdot; \theta_\beta) : \mathcal{X} \to [0, 1]^C$ where $\theta_\beta$ are the parameters of the mapping $\beta$. Equation (1) can be rewritten as follows

$$g(\boldsymbol{B}_I) = \rho([\beta(\boldsymbol{x}_1; \theta_\beta) \cdot f(\boldsymbol{x}_1)], \cdots, [\beta(\boldsymbol{x}_{|I|}; \theta_\beta) \cdot f(\boldsymbol{x}_{|I|})]) \qquad (2)$$

where $\rho$ is an aggregating operator (or the pooling function) and $\theta_\beta$ are the learnable parameters of $\beta$. The weighting function $\beta = \sigma \circ \varphi \circ \phi$ is also modeled as a composition of three functions. $\beta$ shares the feature extractor $\phi$ with the instance classifier $f$, however, applies consecutively $\varphi(\cdot; \theta_\varphi)$ (drawn in green in fig. 1) on the features which outputs an $C$ dimensional unnormalized vector and followed by a univariate Gaussian radial basis normalization $\sigma$ (drawn in blue in fig. 1) defined as follows:

$$\sigma(\boldsymbol{r}; \boldsymbol{\mu}, \boldsymbol{\epsilon}) = \exp(-\|\frac{\boldsymbol{r} - \boldsymbol{\mu}}{\boldsymbol{\epsilon}}\|^2) \qquad (3)$$

where $\boldsymbol{\mu}$ and $\boldsymbol{\epsilon}$ are the mean and standard deviation respectively. We propose two ways of calculating $\boldsymbol{\mu}$ and $\boldsymbol{\epsilon}$. In the first case, we fit $\boldsymbol{\mu}$ and $\boldsymbol{\epsilon}$ to the scores that are extracted over $\boldsymbol{B}_I$, i.e. $\{\varphi(\phi(\boldsymbol{x}_1)), \cdots, \varphi(\phi(\boldsymbol{x}_{|I|}))\}$. $\boldsymbol{\mu}$ and $\boldsymbol{\epsilon}$ are simply statistics and can be calculated by the following formulas:

$$\begin{aligned} \boldsymbol{\mu} &= E_{\{\forall \boldsymbol{x}_i \in \boldsymbol{B}_I\}}[\varphi(\phi(\boldsymbol{x}_i))] \\ \boldsymbol{\epsilon}^2 &= E_{\{\forall \boldsymbol{x}_i \in \boldsymbol{B}_I\}}[\|\varphi(\phi(\boldsymbol{x}_i)) - \boldsymbol{\mu}\|^2] \end{aligned} \qquad (4)$$

In other words, weights can be normalized by using the local bag statistics. These variants are denoted by **GP0T0** and **GP0T1** as explained later in the experiments. Here the intuition is that by comparing each instance with local-bag statistics, our weight learning function is able to learn to detect key instances. In the second one, we set $\boldsymbol{\mu}$ and $\boldsymbol{\epsilon}$ as parameters of our network and jointly optimize them with the rest of the network parameters $(\theta_\phi, \theta_\psi, \theta_\varphi)$. In this case, the parameters are not local to a bag but global, learned from the updates from all the training samples and do not necessarily correspond to a statistical mean and standard deviation of the score distribution anymore. This variant is denoted by **GP1T1**.

**Training.** In this paper we focus on two kinds of MIL tasks, binary and multi-class classification. In the binary case, both the instance classifier $f$ and weighting function $\beta$ output a scalar per instance, their outputs are simply multiplied (see eq. (2)). Afterwards, *log-sum-exponent* [15] is applied over all the instances as the aggregation function $\rho$ and finally, fed into a sigmoid function to get the

probability of being positive for a bag. We use the binary cross entropy loss to train this model. In the multi-class case, both the instance classifier $f$ and weighting function $\beta$ output a $C$ dimensional vector, they are element-wise multiplied, then averaged over the bag and finally normalized by a softmax function over classes. Here we use the cross entropy loss to optimize the model. We train all network parameters jointly by using stochastic gradient descent or Adam. The implementation details are given in section 3.

## 3 EXPERIMENTS

We investigate three variants of our model by varying two design decisions in a controlled manner. First we analyze whether the two stream architecture, specialized streams for instance classification and weighting are beneficial. Second, we study whether the parameters of the proposed Gaussian normalization should be calculated from bag statistics or learned along with the other parameters in the network. We describe three model variants in the following.

**GP0T0.** Here, we set both classifier $\psi(\cdot, \theta_\psi)$ and instance weighting function $\varphi(\cdot, \theta_\varphi)$ to be the same, i.e., $\theta_\psi = \theta_\varphi$. This prevents the network to learn specialized functions for instance classification and weighting and we indicate this by "T0" (two stream off). The Gaussian function variables $\boldsymbol{\mu}$ and $\boldsymbol{\epsilon}$ are estimated from eq. (4) and this is denoted by "GP0" (Gaussian parameters not learnable). Hence, there are no dedicated parameters for learning the weighting function. Since $\boldsymbol{\mu}$ and $\boldsymbol{\epsilon}$ are estimated using within bag instances, prediction weights only depends on local bag information.

**GP0T1** Here we use dedicated parameters for the instance weighting function $\varphi(\cdot, \theta_\varphi)$ and it is realized by a 2-layered neural network with Relu activation. As in the previous case, the parameters of the Gaussian normalization are estimated from eq. (4).

**GP1T1** In this case, we keep the same settings in *GP0T1* but the parameters of the Gaussian normalization ($\boldsymbol{\mu}$ and $\boldsymbol{\epsilon}$) are learned through back-propagation along with the other network parameters. In contrast to the setting where the Gaussian parameters are estimated locally for each bag, we learn a single global parameterization. In the next section we report and analyze our results in different binary and multi-class classification datasets and compare our results to the previous work.

Similar to the instance weighting function $\varphi$, our classifier (i.e. $\phi$) is realized by a 2-layered neural network with Relu activation. The form of feature extractor depends on the problem. We use LeNet model as the feature extractor for CIFAR10 and MNIST datasets. We use a multi layered perceptron (3 fully connected layers with 256, 128 and 64 dimensions, with ReLU activation) for other MIL datasets.

### 3.1 RESULTS FOR BINARY CLASSIFICATION.

First, we evaluate our method on MNIST-bags dataset [10] with the code provided by the authors. A bag is made up of a random number of $28 \times 28$ gray images taken from the MNIST dataset. The number of images in a bag is Gaussian-distributed and a bag is given a positive label if it contains one or more images with the label "9". Following [10], we investigate the influence of the number of bags in the training set and compare all methods. During evaluation we use a fixed number of 1,000 test bags. For all experiments a LeNet5 [12] feature extractor $\phi()$ is used. We use the default training parameters that are used in [10] (mean bag length is 10, variance of the bag length is 2, etc.). All models are trained end-to-end with the same loss function and same amount of epochs. The attention method (**ATT**) introduced in [10] apply Softmax over instance scores returned by an instance classifier and then use those Softmax normalized scores to obtain a bag-level feature representation (or a bag embedding). ATT-G uses the gated mechanism of ATT where a combination of both tanh and sigmoid activations are used [10]. Similarly, **WSDN** [3] method uses a two-stream approach like us, where one stream is used to weight regions proposals using Softmax function and the other stream is used to classify image regions to improve object detection. Finally, in WSDN the sum of weighted region scores are used to train the model. WSDN is originally used for object detection, however here we adapt it for MIL. WSDN is related to this work as it also uses a two-stream approach. Therefore, we compare with these two models using MNIST-bags dataset [10] and results are reported in table 1.

| # of bags | ATT | ATT-G | WSDN | GP0T0 | GP0T1 | GP1T1 |
|---|---|---|---|---|---|---|
| 100 | $0.143 \pm 0.018$ | $0.113 \pm 0.020$ | $0.099 \pm 0.012$ | $0.174 \pm 0.030$ | $0.082 \pm 0.012$ | $\mathbf{0.075 \pm 0.007}$ |
| 200 | $0.063 \pm 0.015$ | $0.070 \pm 0.016$ | $0.072 \pm 0.005$ | $0.100 \pm 0.026$ | $0.059 \pm 0.009$ | $\mathbf{0.053 \pm 0.008}$ |
| 300 | $0.056 \pm 0.014$ | $0.063 \pm 0.003$ | $0.059 \pm 0.027$ | $0.057 \pm 0.015$ | $0.066 \pm 0.015$ | $\mathbf{0.045 \pm 0.010}$ |
| 400 | $0.041 \pm 0.005$ | $0.047 \pm 0.004$ | $0.038 \pm 0.007$ | $0.056 \pm 0.011$ | $0.042 \pm 0.010$ | $\mathbf{0.036 \pm 0.009}$ |
| 500 | $0.036 \pm 0.003$ | $0.040 \pm 0.005$ | $0.037 \pm 0.005$ | $0.053 \pm 0.012$ | $0.041 \pm 0.008$ | $\mathbf{0.033 \pm 0.010}$ |

Table 1: Bag classification performance using average error rate on MNIST-bags dataset [10] by varying the training bas size.

| Bag length | ATT | ATT-G | WSDN | GP1T1 (our) | ATT | WSDN | GP1T1 (our) |
|---|---|---|---|---|---|---|---|
| | Instance detection in AP | | | | Bag recognition in err. rate. | | |
| 10 | $0.644 \pm 0.049$ | $0.658 \pm 0.034$ | $0.788 \pm 0.023$ | $\mathbf{0.931 \pm 0.007}$ | $0.143 \pm 0.018$ | $0.099 \pm 0.012$ | $0.075 \pm 0.007$ |
| 20 | $0.635 \pm 0.062$ | $0.575 \pm 0.116$ | $0.638 \pm 0.123$ | $\mathbf{0.656 \pm 0.051}$ | $0.114 \pm 0.009$ | $0.082 \pm 0.014$ | $0.067 \pm 0.009$ |
| 30 | $0.190 \pm 0.144$ | $0.109 \pm 0.002$ | $\mathbf{0.513 \pm 0.146}$ | $0.467 \pm 0.235$ | $0.053 \pm 0.016$ | $0.048 \pm 0.012$ | $0.039 \pm 0.007$ |

Table 2: Instance detection performance in average precision using MNIST-bags dataset [10] for training size of 100 bags and tested on 1000 bags.

From these results, we see that ATT does not perform as well as WSDN. Gated attention ATT-G, does not perform as good as ATT except in one case. Both ATT and WSDN methods use Softmax function to identify key instances. However, ATT uses Softmax scores for learning an attention-based feature representation for the bag whereas WSDN uses the same information to weight instances. This suggests that perhaps the two stream approach is much effective for multiple instance learning even-though WSDN is initially designed for object detection. Our two stream-off GPOTO does not perform as good as WSDN or the ATT indicating perhaps the two stream approach is need. GP0T1 method which enables two stream approach over Gaussian instance weighting obtains comparable results to ATT and WSDN. This confirms the effectiveness of two-stream approach in multiple instance learning. Finally, we see a significant boost in performance for GP1T1 where we learn the parameters of the Gaussian instance weighting function. The whole idea of attention (weather it is feature attention or instance weighting) is to make use of relative comparisons. Instances that comparatively important are highly weighted to obtain attention weights or instance weights. Softmax-based methods (such as ATT and WSDN) and the Gaussian-based method such as GP0T1 make local comparisons (i.e comparisons within a bag) to find important instances. Interestingly, all of them obtain somewhat comparable results. However, GP1T1 learns the Gaussian parameters using back-propagation and specific to the dataset statistics. Therefore, the instance comparisons are not local to the bag, but learned globally across the dataset. Perhaps this also enforces a regularization effect on the instance representation by making positive instances to form cluster (s) with respect to the instance weighting function. Therefore, we conclude that parametric two-stream-based Gaussian instance weighting is a better strategy for multiple instance learning. Due to it's effectiveness, we use GP1T1 variant of our method in all other experiments.

Now we evaluate the instance detection capacity of each model using MNIST-bags dataset [10]. Specifically, the instance score is obtained by the product of instance weighting and instance classifier in eq. (2), i.e., $\beta(\boldsymbol{x}_k; \theta_\beta) \cdot f(\boldsymbol{x}_k)$. We use similar strategy for WSDN and ATT methods. For ATT [10], as done in that paper, we take the output of the attention module (attention weights returned by Softmax function) as the instance score. Thereafter, by changing the mean length of the bag to 10, 20 and 30, we report the instance detection performance in average precision in table 2. Average precision is used because it is widely used for other detection tasks (e.g. object detection). We also report bag classification performance in error rate as before. As it can be seen, our method obtains better instance detection performance than ATT and WSDN for two different bag lengths. Interestingly, when the bag length is 10, our method is quite accurate obtaining average precision of 0.931 while the second best WSDN obtains only 0.788. When mean bag length is increased to 20, our method still obtains slightly better instance detection performance than both ATT and WSDN. Interestingly, WSDN outperforms both ATT and GP1T1 for instance detection when mean bag-size is 30. Even then, the bag classification of GP1T1 is better than ATT and WSDN. Originally, WSDN is designed for object detection and has the capability to ignore most negative instances and pick only the relevant handful of positives. Therefore, when there are only a few number of positives, the WSDN is able to pick those using Softmax. In contrast, GP1T1 has the capacity to detect more positive instances than Softmax-based WSDN. Therefore, GP1T1 has the capacity to obtain better MIL recognition performance as we will see next.

| Method | MUSK 1 | MUSK 2 | Fox | Tiger | Elephant |
|---|---|---|---|---|---|
| | Traditional methods. | | | | |
| mi-SVM [1] | $0.874 \pm$ N/A | $0.836 \pm$ N/A | $0.582 \pm$ N/A | $0.784 \pm$ N/A | $0.822 \pm$ N/A |
| MI-SVM [1] | $0.779 \pm$ N/A | $0.843 \pm$ N/A | $0.578 \pm$ N/A | $0.840 \pm$ N/A | $0.843 \pm$ N/A |
| MI-Kernel [9] | $0.880 \pm 0.031$ | $0.893 \pm 0.015$ | $0.603 \pm 0.028$ | $0.842 \pm 0.010$ | $0.843 \pm 0.016$ |
| EM-DD [20] | $0.849 \pm 0.044$ | $0.869 \pm 0.048$ | $0.609 \pm 0.045$ | $0.730 \pm 0.043$ | $0.771 \pm 0.043$ |
| mi-Graph [21] | $0.889 \pm 0.033$ | $0.903 \pm 0.039$ | $0.620 \pm 0.044$ | $0.860 \pm 0.037$ | $0.869 \pm 0.035$ |
| miVLAD [17] | $0.871 \pm 0.043$ | $0.872 \pm 0.042$ | $0.620 \pm 0.044$ | $0.811 \pm 0.039$ | $0.850 \pm 0.036$ |
| miFV [17] | $0.909 \pm 0.040$ | $0.884 \pm 0.042$ | $0.621 \pm 0.049$ | $0.813 \pm 0.037$ | $0.852 \pm 0.036$ |
| | Deep learning methods. | | | | |
| mi-Net [16] | $0.889 \pm 0.039$ | $0.858 \pm 0.049$ | $0.613 \pm 0.035$ | $0.824 \pm 0.034$ | $0.858 \pm 0.037$ |
| MI-Net | $0.887 \pm 0.041$ | $0.859 \pm 0.046$ | $0.622 \pm 0.038$ | $0.830 \pm 0.032$ | $0.862 \pm 0.034$ |
| MI-Net DS [16] | $0.894 \pm 0.042$ | $0.874 \pm 0.043$ | $0.630 \pm 0.037$ | $\mathbf{0.845 \pm 0.039}$ | $\mathbf{0.872 \pm 0.032}$ |
| MI-Net RC [16] | $0.898 \pm 0.043$ | $0.873 \pm 0.044$ | $0.619 \pm 0.047$ | $0.836 \pm 0.037$ | $0.857 \pm 0.040$ |
| WSDN[3] | $0.850 \pm 0.011$ | $0.822 \pm 0.012$ | $0.512 \pm 0.012$ | $0.794 \pm 0.010$ | $0.828 \pm 0.008$ |
| ATT [10] | $0.892 \pm 0.040$ | $0.858 \pm 0.048$ | $0.615 \pm 0.043$ | $0.839 \pm 0.022$ | $0.868 \pm 0.022$ |
| Gated-ATT [10] | $0.900 \pm 0.050$ | $0.863 \pm 0.042$ | $0.603 \pm 0.029$ | $\mathbf{0.845 \pm 0.018}$ | $0.857 \pm 0.027$ |
| GP1T1 (Ours) | $\mathbf{0.911 \pm 0.040}$ | $\mathbf{0.902 \pm 0.047}$ | $\mathbf{0.646 \pm 0.054}$ | $0.834 \pm 0.081$ | $0.822 \pm 0.064$ |
| | Our best result with different pooling. | | | | |
| GP1T1 (Ours)-Best | $\mathbf{0.923 \pm 0.009}$ | $\mathbf{0.907 \pm 0.009}$ | $\mathbf{0.646 \pm 0.054}$ | $\mathbf{0.874 \pm 0.008}$ | $0.847 \pm 0.009$ |

Table 3: Average classification accuracy ($\pm$ standard deviation) on MUSK 1, MUSK 2, Fox, Tiger and Elephant datasets. We directly compare with Deep learning methods as they are the most relevant for us. Results from traditional methods are shown as a reference. We bold results only for best performing deep learning methods.

Now, we evaluate on five historically important binary MIL benchmarks, MUSK 1-2 [7], Fox, Tiger and Elephant image datasets [1]. All five datasets provide only precomputed features and small number of bags. MUSK1 and MUSK2 datasets present a real-world problem in drug activity analysis. The goal is to predict activity of new molecules without requiring expensive process of synthesizing the most promising candidate molecules which can yield large economical savings. Both MUSK1 and MUSK2, consist of descriptions of molecules using multiple low-energy conformations. Each conformation is represented by a 166-dimensional feature vector derived from surface properties. MUSK1 contains on average approximately 6 conformation per molecule, while MUSK2 has on average more than 60 conformations in each bag [1]. Fox, Tiger and Elephant datasets contain features computed over image segments by using region descriptors. Each positive bag consists at least one patch from the target animal category. Negative ones consists patches from other animals. Feature dimensions are 166 for MUSK1 and MUSK2 and 230 for the remaining three. The standard performance measure is average classification accuracies (and their standard deviation) over five ten-fold validation runs. We follow the experimental setting in [10, 16] and use the same architecture (3 fully connected layers with 256, 128 and 64 dimensions, ReLU) as the feature extractor $\psi$ and use SGD optimizer with batch size 1 as in [10]. We use exactly the same hyper parameters used in prior work [10, 16]. Results are reported in table 3.

Interestingly, our method is able to outperform some of the traditional as well as recent deep learning methods on three datasets, MUSK1, MUSK2 and more challenging Fox dataset. The results show that our GP1T1 variant with learnable Gaussian parameters with a two stream architecture is extremely effective even for traditional multiple instance learning tasks. Most interestingly, our method outperforms both ATT and WSDN in most cases.

So far, we use the log-sum-exponent [15] as the aggregation function $\rho$ in eq. (2). Here we investigate the classification performance of various pooling operators $\rho()$ in eq. (2) and report results in table 4. Interestingly, mean pooling is as effective as the log-sum-exponent on these traditional datasets. Our method is able to obtain quite good results on four datasets irrespective of the pooling operator except for Elephant dataset. Max operator seems the least effective one, perhaps due to lack of gradient information. We believe that log-sum-exponent is theoretically more relevant for MIL and therefore we recommend using that even the mean pooling seems to obtain better results on this dataset. So far we have demonstrated the effectiveness of our method on binary multiple instance learning problem for end-to-end learning (tables 1 and 2) and non-end-to-end learning (tables 3 and 4).

## 3.2 Results for multi-class classification

Here we introduce a new dataset that contains raw input (in contrast to precomputed ones in the previous experiments) and also multiple categories. To evaluate our method in multi-class classification

| $\rho()$ | MUSK 1 | MUSK 2 | Fox | Tiger | Elephant |
|---|---|---|---|---|---|
| Log-Sum-Exp. | **0.911 ± 0.040** | **0.902 ± 0.047** | **0.646 ± 0.054** | 0.834 ± 0.081 | 0.822 ± 0.063 |
| Max | 0.880 ± 0.012 | 0.868 ± 0.011 | 0.529 ± 0.013 | 0.805 ± 0.012 | 0.790 ± 0.017 |
| Mean | **0.919 ± 0.009** | **0.907 ± 0.009** | 0.614 ± 0.012 | **0.874 ± 0.008** | 0.840 ± 0.011 |
| Sum | **0.923 ± 0.009** | 0.797 ± 0.015 | 0.615 ± 0.012 | **0.862 ± 0.008** | 0.847 ± 0.009 |

Table 4: Comparing pooling operators $\rho()$ for our method (GP1T1) on five MIL standard dataset. Results better than state-of-the-art deep learning methods are in bold and the best pooling operator is underlined for each dataset.

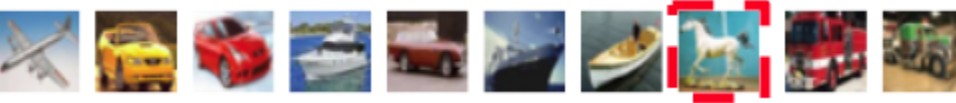

Figure 2: A bag from CIFAR-10-MIL dataset with one positive instance (horse) where the bag size is 10. The bag is labeled as category horse. Each bag has positives only from a single category.

task, we create a new multiple-class MIL dataset from CIFAR-10 dataset [11] and call it *CIFAR-10-MIL dataset*. We treat six natural object classes ("bird", "cat", "deer", "dog", "frog", "horse") as positive instances and rest of the object categories as negatives. Each bag contains multiple negative instances from four negative classes ("automobile", "truck", "ship", "air-plane") while positive instances comes from a single positive class. We generate 30,000 and 6,000 bags for training and evaluation respectively. We will release the code for generating CIFAR-10-MIL dataset along with all other codes used for experimentation. An example bag is depicted in fig. 2. The objective is to predict the bag labels accurately.

Bag generation procedure involves three parameters: bag size, minimum and maximum number of positive instances in a bag. This dataset allows us to change the ratio of positive/negative instances in bags by varying them and study the effect of ambiguity in this task. For instance, in case of a bag of size 100 with maximum of 5 positive instances and a minimum of 1, the ambiguity could vary from 95% to 99%. Thus controlling the positive/negative ratio allows us to evaluate how MIL algorithms deals with certain amount of ambiguity levels within the bag. In all these experiments, we build our model on LeNet5 model [12] and use it as the feature extractor which outputs 84 dimensional feature vector for each image (instance) within the bag. We train all the reported models end-to-end by using stochastic gradient descent with a training batch size of 128 bags[1]. The learning rate is set to 0.1 for 150 epochs and lowered to 0.01 and 0.001 after 50 epochs at each time.

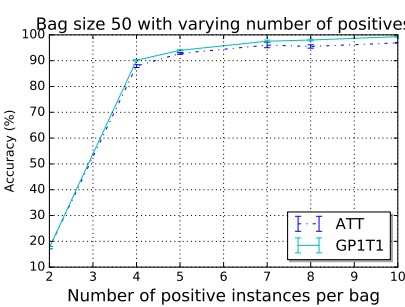

Figure 3: Results for varying the number of positive instances for a fixed bag size of 50.

In the first experiment, we use a fixed bag size of 50, sample exactly $N$ number of positive instances for each bag and vary it from 2 to 10. In other words, we vary the amount of ambiguity in the bags used for training. We compare our method (GP1T1) to the recent state-of-the art method ATT [10]. Note that we use same bag size for both train and test in this experiment. Results are depicted at multiple $N$ values in fig. 3.

Both methods perform better in bag classification as the number of positives increases and ambiguity decreases. Interestingly, both methods fail to learn in the presence of extreme ambiguous level of 96% (only 2 positive instances in a bag of size 50). Both methods recover (learn), when the ambiguous level is reduced to 92%. GP1T1 method outperforms ATT for all ambiguous levels indicating that GP1T1 is better than ATT even for multiple class multiple instance learning.

---

[1]We also experimented with Adam and obtained similar results.

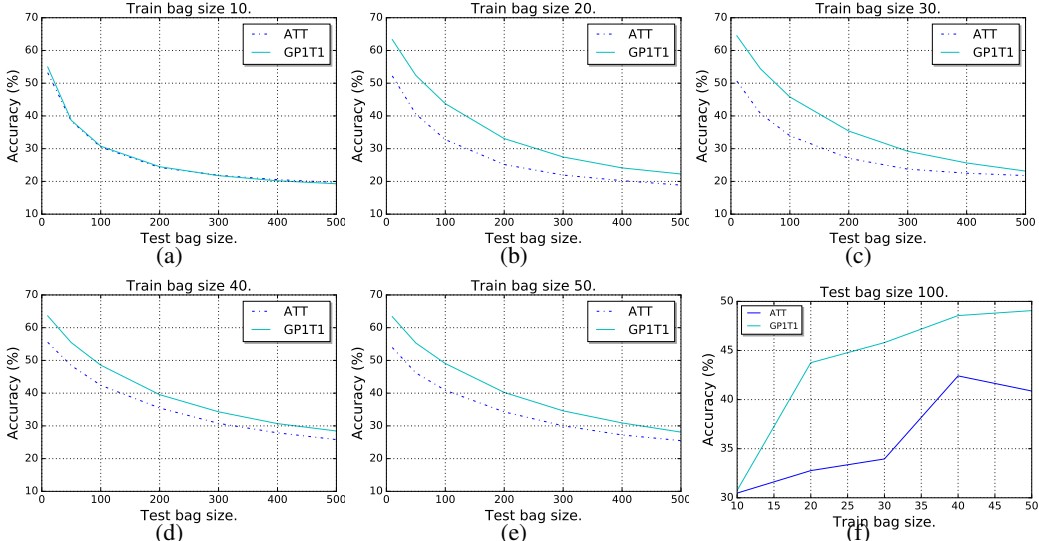

Figure 4: (a-e): Multi-class MIL classification performance (averaged over 10 runs) on CIFAR-10-MIL dataset. Each training bag has a maximum of 10 positive instances and a minimum of 1. Training bag size varies from 10 to 50 and performance is evaluated on test bags having exactly one positive instance. Test bag size varies from 10, 50, 100, 200, 300, 400, 500. (f): We test models on bags of size 100 having one positive instance. We vary the training bag size from 10 to 50 having maximum of 10 positive instances and minimum of one.

### 3.3 Robustness of end-to-end learned MIL methods.

In this section, we further study the robustness characteristics of the presented methods when the ambiguity levels in train and test bags differ. To this end, we first train each method with fixed size bags, each containing 1 to 10 positives and then evaluate them on varying bag sizes, each containing only 1 positive. The average classification accuracies of the models are depicted in fig. 4. In each subplot (a)-(e) we fix the training bag size to a certain number (10, 20, 30, 40, 50) and depict the performance of each method for varying test bag sizes. In fig. 4.(f) we fix the test bag size to 100 and show the results for the models that are trained with different bag sizes. Based on the results, we expect to answer the following first two questions.

**(i) How robust are the presented MIL methods against extremely ambiguous test bags?**
As we can see from fig. 4.(a)-(e), our GP1T1 method consistently outperform ATT method across various train and test bag sizes. Interestingly, when the train bag size is 10, both methods perform equally. Nevertheless, when the train bag size increases, our method stats to perform better than ATT indicating that ours can better learn under more ambiguous bags. However, both methods perform poorly as the test bag size (and ambiguity) increases. In particular when the models are first trained on less ambiguous bags such as bag size of 10 fig. 4.(a), even though the performance for test bag of size 10 is quite good, the performance of a test bag size of 500 is very poor ( 20%). However, when the models are trained with even more ambiguous bags (fig. 4 (e)), they perform better for larger test bags (e.g. 500) where the performance is around 30% for GP1T1. Interestingly, our models are better than the attention based models when tested with very large bags with more ambiguous bags. This indicates the robustness of our methods to varying amount of ambiguity (see fig. 4 (e)).

**(ii) How transferable are the MIL models with respect to change in ambiguity levels?**
Here we analyze the limits of training bag ambiguity that the MIL model can handle reasonably. The results for the study are depicted in fig. 4.(f). First we observe that all the models benefit from increase in the ambiguity level up to bag size of 40 and then the performance of ATT [10] degrades. However, our model GP1T1 retains its performance. Here we hypothesize that the models which can learn instance weighting accurately are more robust to changes in ambiguity ratios.

## 4 Conclusion

Our deep MIL method uses a novel two stream method for instance weighting and classification to compute the bag prediction score. We evaluate our method in both binary and multi-class classification tasks in several standard MIL benchmarks and obtain excellent results for both. We show that presence of extreme ambiguity in training bags (low positive/negative instance ratio) hurts the performance of the MIL methods severely. However, we also demonstrate that when the MIL methods are successfully trained with bags containing moderate ambiguity, most methods would also be able to correctly classify bags even in case of extreme ambiguity. Our method which uses a Gaussian function with dedicated learnable parameters, is able to outperform other methods under extreme presence of ambiguity while obtaining better results on traditional MIL datasets such as MUSK. Interestingly, in most cases, our method can also obtain good instance detection results.

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

# A  APPENDIX

## A.1  LARGE SCALE UNTRIMMED VIDEO CLASSIFICATION WITH MIL.

In this section we perform a very large scale untrimmed video action classification using four MIL approaches and show that 1. MIL in general is useful even for a large scale video classification problem 2. our method still outperforms other competitors on this task. We select ActivityNet 1.3 video dataset which consists of 10,000 video for training and about 5000 videos for validation [8]. We consider 32 frame video clip as an instance and the entire video as the bag. A video in ActivityNet1.3 may have one or more human actions. Our objective is to recognize the presence and absence of each action category in each video. Therefore, the performance is measured using mean average precision. We train all methods using I3D network as the feature extractor [5]. We use batch size of 64 videos and Adam optimizer with learning rate of 0.0005. We use a weight decay of $10e^{-5}$. All models are trained with binary cross entropy loss. We report results in Table 5.

| Method | mean AP |
|---|---|
| I3D Baseline | 0.7512 |
| ATT | 0.7605 |
| ATT-G | 0.7768 |
| WSDN | 0.7864 |
| GP1T1 | **0.8223** |

Table 5: Action classification performance on ActivityNet1.3 dataset using MIL methods.

## A.2  ARE THE INSTANCE WEIGHTS MEANINGFUL?

Here we provide some experimental results for instance detection weights using Cifar10-MIL dataset. We empirically verify that positive instances are weighted higher than the negative ones by $\beta$. For GP1T1 method positive vs negative weights returned by $\beta$ are 0.94 $\pm$ 0.11 vs. 0.54 $\pm$ 0.10 respectively. We conclude, our method learns meaningful weights for instances and can robustly detect positive instances under different test distributions from the results shown in table 2.

