# OpenReview forum: "Deep Multiple Instance Learning with Gaussian Weighting"
_ICLR.cc/2020/Conference — Reject_

### Official Review · AnonReviewer1 · 2019-10-23
**Official Blind Review #1**

**Rating:** 8

**Review:**

The paper proposes a new approach to weighting in multiple-instance learning scenario. They multiply scores for instances by Gaussian normalized weights. The hyperparameters of the Gaussian RBF are either estimated locally (i.e., within a bag) or trained across bags. Additionally, the authors verify whether it is better to have two separate neural networks (one for instance scoring and one for instance weighting) or share weights between them. Eventually, a variant with globally trained hyperparameters of the Gaussian RBF and separate weights of neural networks performs the best.

In general, the paper is very well written and easy to follow. The proposed solution is reasonable and, importantly, performs better than other SOTA approaches. The organization of the paper is proper, all concepts are well explained. The experiments are meaningful and answer important questions (i.e., which of the considered variants is the best, how the proposed approach compares to SOTA methods). In my opinion, the paper could be accepted.

REMARKS:
- What is the temperature value used in the log-sum-exp?

- The authors claim that the log-sum-exp pooling operator is theoretically more relevant for MIL. Could you comment more on that?

- Why do the authors compare to ATT instead of Gated-ATT? In [8] it was shown that Gated-ATT performs better.

======== AFTER REBUTTAL ========
I would like to thank the authors for their rebuttal. I appreciate their answers and updates. I still believe that the paper is interesting and important for the MIL community. I sustain my decision.

**Experience Assessment:**

I have published one or two papers in this area.

**Review Assessment: Checking Correctness Of Derivations And Theory:**

I assessed the sensibility of the derivations and theory.

**Review Assessment: Checking Correctness Of Experiments:**

I assessed the sensibility of the experiments.

**Review Assessment: Thoroughness In Paper Reading:**

I read the paper at least twice and used my best judgement in assessing the paper.

---

> ### Author Response · Authors · 2019-11-14
> **Clarifications for Official Blind Review #1**
>
> Thanks for the very encouraging comments and feedback. We highly appreciate your remarks.
>
> Q. What is the temperature value used in the log-sum-exp?
> We have evaluated the performance with several temperature values. We observed that small gains are possible when temperature is tuned for each dataset and model. But for simplicity and preventing overturning our model, we have selected to use a global temperature of 1.0 for all settings.
>
> Q. Why log-sum-exp good for MIL?
> Log-sum-exp is a smooth approximation to the maximum function. In contrast to the maximum operator, log-sum-exp function is continuous and its gradient is dense. Thus it is less likely prone to “stick” to a single instance in each bag and to a local minimum during training.
>
> Q. Why do the authors compare to ATT instead of Gated-ATT?
> In our experiments, Gated-ATT only obtained slightly better performance or even less performance than ATT in some cases. At the same time the training time increases. Therefore, we opted to use ATT instead of Gated-ATT. However, for completeness, we now report results for Gated-ATT in Table 1, Table 2 and Table 3.

---

### Official Review · AnonReviewer3 · 2019-10-23
**Official Blind Review #3**

**Rating:** 3

**Review:**

The paper describes a deep multiple instance learning method. It uses
two stream neural networks, one for instance classification and one
for learning normalized weights. The proposed method can be used for
binary and multi-class bag classification.

The authors experimentally show the performance of their proposed
method on several datasets and compare against other state-of-the-art
methods. Also they created a new data set for multi-class bag
classification.

The results are very convincing but the description of the method is
not very clear. In particular, the description of the instance
classifier is confusing. It says that "f" is the instance classifier,
which is a composition of a feature extractor (\phi) and a classifier
(\psi), both modeled as deep neural networks. What are those neural
networks?

Also the weighting function is a composition of three functions, the
Gaussian normalization, the instance weighting, and the feature
extractor. Again the instance weighting process, which is another key
feature of the paper, is not clearly explained.

The paper has several English errors which makes sometimes difficult
to follow all the ideas.


**Experience Assessment:**

I do not know much about this area.

**Review Assessment: Checking Correctness Of Derivations And Theory:**

I assessed the sensibility of the derivations and theory.

**Review Assessment: Checking Correctness Of Experiments:**

I assessed the sensibility of the experiments.

**Review Assessment: Thoroughness In Paper Reading:**

I read the paper at least twice and used my best judgement in assessing the paper.

---

> ### Author Response · Authors · 2019-11-14
> **Clarifications to Official Blind Review #3**
>
> Thanks for the encouraging comments. Indeed, our method obtains very strong results on both classical and modern MIL datasets and tasks. We also add a new large scale video classification task (648 hours of video). Our method performs better than others even on this large scale problem. See the supplementary section.
>
> Q. Architecture of networks
> - The feature extractor (\phi) is dataset specific. It is the LeNet5 model for the CIFAR10 and MNIST Bags experiments, where input samples are images.
> - For other MIL datasets, where each input sample is a vector, the feature extractor is chosen to be a 3 layer MLP (256-128-64 dimensions, each followed by ReLU). We add this clarification at the beginning of the  experiments section in Blue color for clarity.
> - Both the instance weighting function (\varphi) , instance classifier function (\psi) are 2-layer MLP networks with a ReLU activation between the layers. The instance weighting process is depicted in Equation 2. The instance weighting \beta(x) is multiplied with instance score f(x) and aggregated with pooling function \rho. To make it clearer, we also added the square bracket []. We also believe that figure 2 explains how overall all components work together. Any suggestions to improve figure 2 is highly appreciated
>
> In the new version of the paper, We tried our best to make the language of the paper smoother. Any specific comment is highly appreciated.

---

### Official Review · AnonReviewer2 · 2019-10-28
**Official Blind Review #2**

**Rating:** 3

**Review:**

This paper described an approach of performing multiple instance learning (MIL) by using a network branch to weight instances and then using a Gaussian normalization layer on top of it, where the weightings are predicted based on in-bag variances. The instance weighting scheme, a classic in MIL, has been proposed in deep networks by [3]. Hence, I don't see much novelty here except that there is a Gaussian normalization layer after the instance weighting in the MIL framework.

I'm a bit worried that sigma seems to be unnormalized in the first case -- what would happen if the bag score distribution is non-Gaussian? GP1T1 seems more sensible by learning all these weights.

However, I see significant issues in terms of evaluation which makes hard to accept this paper.

I firmly believe that 22 years after the (Dietterich 1997) paper, it's no longer enough to only use the original MIL datasets and classification datasets such as the CIFAR-10 bags to evaluate MIL. I may be alone here which is open to discussions, but the original motivation for MIL is for the problem to be weakly-supervised where we only know a high-level label but no low-level labels. There exist many realistic image problems that are similar to this (e.g. weakly-supervised detection and semantic segmentation) and have received a plethora of work, hence it's unclear to me whether still using these arbitrarily generated CIFAR-10 bags and the 5 old datasets would still apply to MIL as an approach. After all, MIL is almost dying and "weakly-supervised learning" has risen in popularity with almost being the same problem as MIL. I think for MIL to work its way back, it should first start by using the right datasets to test (e.g. the datasets in [3] would be a great starting point) and comparing with other weakly-supervised detection approaches that do not use additional information.

Besides the philosophical point, a practical issue is that the numbers seem to be bold arbitrarily according to the authors' whim. In table 3, MI-Net DS is better than every other method in the last 2 columns, but not bold, and also it seems that no t-test was performed to determine the significance of differences. This also happens in Table 1, where I think WSDN should be equivalent with the proposed method in the last 2 rows.

Finally, the choice of excluding MI-Net in the CIFAR experiments is dubious as well, given its performance in the simple datasets. Why is MI-Net not tested on the CIFAR experiments?

**Experience Assessment:**

I have published one or two papers in this area.

**Review Assessment: Checking Correctness Of Derivations And Theory:**

I assessed the sensibility of the derivations and theory.

**Review Assessment: Checking Correctness Of Experiments:**

I carefully checked the experiments.

**Review Assessment: Thoroughness In Paper Reading:**

I read the paper at least twice and used my best judgement in assessing the paper.

---

> ### Author Response · Authors · 2019-11-14
> **Clarifications to Blind Review #2**
>
> We thank the reviewer for the valuable feedback and address the questions below.
>
> Q) Novelty over WSDN [3]
>
> Our paper is indeed reminiscent of two stream architectures such as WSDN [3] and Contextlocnet by Kantorov et al., however, it differs in two fundamental aspects:
>
> First, the two stream architecture is originally designed for weakly supervised object detection (instance classification). Here our focus is learning to classify bags of instances rather than individual ones. One may argue that  achieving perfect instance classification would also yield perfect classification result at bag level. However, in practice (in the presence of noise), optimizing for the former does not guarantee optimizing for the latter. As a matter of fact, our paper systematically analyses the gap between these two tasks and also their robustness to various noise levels.
>
> Second our method uses Gaussian normalization, where in-fact the parameters of the RBF function can be learned and is more representative than softmax function. Gussian parameters can be learned per-class whereas in the soft-max normalization, there is no such flexibility. Our hypothesis is that different classes have different statistics and the Gaussian layer introduced in our method can handle this. Though these layers are conceptually simple, we show that these layers are essential for good performance and they yield significant improvement over WSDN. See also our new results on ActivityNet1.3 for multi-class, multi-label video classification with MIL.
>
> Q) I'm a bit worried that sigma seems to be unnormalized in the first case -- what would happen if the bag score distribution is non-Gaussian? GP1T1 seems more sensible by learning all these weights.
>
> We agree with the reviewer that actually the GP1T1 is the most robust method and can adapt the parameters to the data. It is possible that the bag scores are non Guassian and in this case other variants will fail. However, GP1T1 variant is more flexible and has the capacity to adapt and modify the representation.
>
> Q) it's no longer enough to only use the original MIL datasets and classification datasets such as the CIFAR-10 bags to evaluate MIL, … start by using the right datasets to test (e.g. the datasets in [3])
> We disagree with the reviewer due to two reasons. First, weakly supervised learning focuses on instance classification and thus its datasets must contain instance labels for evaluating the final performance. Whereas the traditional MIL datasets does not have such limitation, eg in the MUSK dataset, instance labels are not available, as it is not possible to know which conformations (instance-level) are musk or not musk but only which molecules (bag-level) are. Another  common MIL task, where instance labels are not present, is sentiment analysis. Individual sentences would not directly convey any sentiment, but as a whole, the entire review may contribute to the sentiment.
>
> Therefore, Multiple instance learning is not a replacement for weakly supervised learning or vice versa. In weakly supervised learning, the objective is to get a good performance for instance recognition. However, there are some problems that instance recognition is impossible or does not make sense. This is why the multiple instance learning was introduced in the first place. It is true that some multiple instance learning methods can be adopted to solve weakly supervised learning methods. Secondly, multiple instance learning datasets such as CIFAR10-MIL and MNIST-MIL-Bags dataset provide a testbed for extensive analysis on the properties of the algorithms and methods. One can vary the bag-size, positive-negative ratio, and evaluate the impact of these methods on these variable factors. Traditional weakly supervised image datasets such as PASCAL-VOC, MS-COCO does not support these kind of analysis. There is no way of controlling the number of  positive and negative instances here or the bag size. Therefore, CIFAR10 and MNIST-BAGS dataset is excellent for these analysis. We cleary disagree with the point that MIL is dying.
>
> Nevertheless, we also perform a large scale untrimmed video classification experiment on ActivityNet1.3. We report mean average precision over 200 action classes. We compare WSDN, ATT, ATT-Gated with our GP1T1 method. We include more information  in the revised paper in the supplementary results section.
> Results are as follows:
> Baseline I3D : 0.7512
> ATT : 0.7605
> Gated ATT : 0.7768
> WSDN: 0.7864
> GPIT1 (Our): 0.8223
>
> Thanks for pointing out the missing bold point. We made the corrections as per your suggestions in the revised paper.
>
> Q) Why is MI-Net not tested on the CIFAR experiments?
> MI-NET is not evaluated on CIFAR10 dataset as it was not performing as good as ATT method and ATT method is the best competitor for our method. Besides, on small dataset our method outperforms the MI-Net method by a very large margin on four cases and MI-Net only obtains better results only in one case.

---

### Decision · Program_Chairs · 2019-12-19

**Decision:**

Reject

**Comment:**

The authors propose a novel MIL method that uses a novel approach to normalize the instance weights. The majority of reviewers found the paper lacking in novelty and sufficient experimental performance evidence.